# LinkNER: Linking Local Named Entity Recognition Models to Large Language Models using Uncertainty

Submission Id: 563*

## ABSTRACT

Named Entity Recognition (NER) serves as a fundamental task in natural language understanding, bearing direct implications for web content analysis, search engines, and information retrieval systems. Fine-tuned NER models exhibit satisfactory performance on standard NER benchmarks. However, due to limited fine-tuning data and *lack of knowledge*, it performs poorly on unseen entity recognition. As a result, the usability and reliability of NER models in web-related applications are compromised. Instead, Large Language Models (LLMs) like GPT-4 possess extensive external knowledge, but research indicates that they *lack specialty* for NER tasks. Furthermore, non-public and large-scale weights make tuning LLMs difficult. To address these challenges, we propose a framework that combines small fine-tuned models with LLMs (LinkNER) and an uncertainty-based linking strategy called RDC that enables fine-tuned models to complement black-box LLMs, achieving better performance. We conduct experiments on standard NER test sets as well as noisy social media datasets. We find that LinkNER can improve performance on NER tasks, especially outperforming SOTA models in challenging robustness tests (with a 3.04% ∼ 21.30% improvement in the F1 score). Additionally, we conduct a quantitative study to examine the impact of key components, such as uncertainty estimation methods, LLMs, and in-context learning, on various NER tasks and provide targeted web-related recommendations.

## CCS CONCEPTS

• **Computing methodologies → Information extraction**.

## KEYWORDS

Information extraction, uncertainty estimation, robustness, large language models

**ACM Reference Format:**
Anonymous Author(s). 2024. LinkNER: Linking Local Named Entity Recognition Models to Large Language Models using Uncertainty. In *Proceedings of In Proceedings of The Web Conference 2024 (WWW '24)*. ACM, New York, NY, USA, 12 pages. https://doi.org/XXXXXXX.XXXXXXX

## 1 INTRODUCTION

Named entity recognition (NER) is a core information extraction task in natural language processing (NLP), where named entities

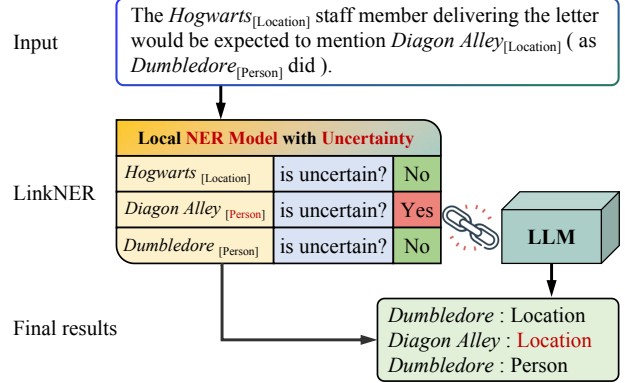

**Figure 1: Illustration of LinkNER processing the NER task. If the local NER model is uncertain about its output (see the second case), the linked LLM undergoes further classification to determine the final result.**

(NEs) are human-defined words or phrases whose entity types are associated with contextual semantics. For instance, in the sentence *"New York never sleeps, a city teeming with diversity"*, the phrase *"New York"* is defined as the "Location" entity. NER models require accurate entity boundary detection and entity type classification for correct recognition. NER extracts structured information from unstructured data, enabling more effective information retrieval and analysis. Therefore, it plays a crucial role in various domains such as web search [14], relation extraction [30], and conversational agents [41]. With the emergence of pre-trained models (PLMs, e.g. BERT [8]), fine-tuning models based on PLMs are suitable for various NLP tasks, which also promotes significant progress in NER tasks.

Recent studies show that NER models face challenges in predicting entities that are not encountered during training [11, 25, 34], encompassing both Out-of-Vocabulary (OOV) and Out-of-Domain (OOD) entities. These challenges frequently arise in web social media and the medical domain [7, 18]. Consequently, NER models' ability to detect and classify decreases when they encounter these entities, which we attribute to the *"lack of knowledge"* of fine-tuned small models. An effective technique, as proposed by Zhang and Yang [39], involves addressing OOV/OOD issues by integrating entity vocabulary through the incorporation of external knowledge. However, it is worth noting that external knowledge may not always be readily accessible or at hand. Strategies for addressing this challenge also involve methods that consider data distributions [34, 40], but these methods still suffer from drastic performance gaps on various disturbing unseen entities (even the SOTA method achieves only a 54.86% F1 score on the social media dataset WNUT'17 [7] with unseen entities). Therefore, current

**Table 1: Performance comparison of GPT-3.5 and SOTA methods on NER dataset. The evaluation metric is the F1 score (%), and Ratio@SOTA$\in [0\%, 100\%]$ refers to the extent of GPT-3.5 reaching the SOTA score.**

| Setting | CoNLL'03 | Onto. 5.0 | JNLPBA |
|---|---|---|---|
| GPT-3.5 | 67.08 | 51.15 | 41.25 |
| Ratio@SOTA | 71.36% | 55.90% | 52.91% |

NER models obviously fail to meet the requirements for system availability and reliability in an open environment.

The advent of large language models (LLMs), such as GPT-3 [4], Bloom [27] and Llama 2 [33], has brought about a turning point for worldwide artificial intelligence, and led to significant progress in a wide range of NLP tasks. With their vast search space and extensive training data, LLMs possess a huge amount of knowledge and have the potential to address OOV/OOD entities. However, while they excel at understanding context and conversational language generation, LLMs are less performed in NER. Recent evaluation indicates that there is a significant performance gap between LLMs and SOTA NER methods (shown in Table 1), though LLMs adopt a few in-context samples as prompts [16]. This may be caused by a lack of specific learning and explicit understanding of the NER task. We attribute this limitation to the LLM's *"lack of specialty"*. Furthermore, LLMs have a substantial number of parameters, some of which remain undisclosed. This presents significant challenges when fine-tuning them to achieve the *"specialty"* of NER. Thus, directly tackling NER tasks using NER models or LLMs alone remains a challenging endeavor.

To address these challenges, we propose LinkNER, shown in Figure 1, a novel approach that enables a local NER model to link synergistically with a black-box large language model (GPT-3.5 and Llama 2-Chat are used in this study), thus making the recognition of various entities more robust. Specifically, we first fine-tune a local NER model. Generally, when equipped with an uncertainty estimation method, the local model can assume any architecture. In this study, we select the commonly used SpanNER framework to equip four uncertainty estimation techniques (based on Confidence, Entropy, Monte-Carlo Dropout [12], and Evidential-based learning [40]) for experiments and analysis. As a result, the local model can recognize simple entities, and detect difficult and unseen entities by assigning them higher uncertainty.

Then, during the inference phase, LLM is employed to further classify the difficult entities. Uncertainty serves as the catalyst for LLM to undergo a paradigm shift in NER tasks, transitioning from entity recognition to entity classification. The final prediction combines the results of the local model and LLM. Consequently, the remarkable generalization ability of LLM overcomes the challenge of *lack of knowledge* in local models. At the same time, the distribution learned by local models addresses the *lack of specialty* challenge faced by LLM, making the two models mutually complementary. To evaluate the effectiveness of LinkNER, we conduct extensive experiments on various datasets and achieve impressive results, and gave recommendations for different LinkNER components on web-related scenarios.

Our contributions are summarized as follows:

- To the best of our knowledge, LinkNER is the first work exploring how to fine-tune models synergistically with LLMs for NER tasks.
- We propose a linking strategy RDC based on uncertainty estimation, which guarantees that two models can complement each other, so as to solve the challenges of *"lack of knowledge"* in fine-tuning small models, and *"lack of specialty"* in LLMs.
- We conduct extensive experiments on LinkNER's benchmarks and robustness tests, and it is notably superior to the current SOTA in robustness tests such as web social media and medical. Moreover, we study the impact of uncertainty estimation methods, LLMs, and in-context learning on LinkNER, while also providing application suggestions.

## 2 RELATED WORK

**Named Entity Recognition**. In web search, where NEs can be used to filter or re-rank to the top. For example, Guo et al. [15] use query log data and latent dirichlet allocation to adopt a probabilistic method to complete the NER query task. Fetahu et al. [9] enhance existing multilingual Transformers based on external dictionary knowledge so that they can simultaneously distinguish and process entity and non-entity query terms in multiple languages.

Recently, NER systems are undergoing a paradigm shift [10, 23, 34], in which the span-based performs better on sentences with OOV words and entities with medium length [10]. In the traditional sequence labeling paradigm, Li et al. [22] propose a boundary-aware bidirectional neural network to solve problems such as boundary label sparsity. Recent studies [1, 25, 34] indicate that test entities present in the training set consistently perform better, while unseen entities lead to lower performance. The current strategy incorporates external knowledge to reduce reliance on word embeddings [21, 39], enabling models to make more semantic-based decisions. Meanwhile, some works focus on distribution-based strategies to solve unseen entity recognition. For example, MINER [34] utilizes mutual information to handle OOV NER tasks and achieve high performance. Additionally, several studies focus on uncertainty-based unseen entity detection [20, 40]. For example, Lei et al. [20] utilize deep learning combined with evidence theory [19] to achieve entity uncertainty estimation. Zhang et al. [40] combine uncertainty-guided reweighting and regularization techniques with evidence theory to improve the NER system's ability to detect unseen entities.

**Language Model Plug-ins**. LLMs currently demonstrate the ability to learn contextually, and external tools also enhance their capabilities [4]. One example is HuggingGPT [31] utilizes ChatGPT for task planning and selects a model based on the available feature description in Hugging Face. It executes each subtask using the selected AI model and aggregates responses based on the execution results. Toolformer [28] introduces special symbols that allow LLMs to call external APIs and accomplish tasks. In the NER tasks, Han et al. [16] observe that GPT-3.5 lacks sensitivity to entity order and struggles to accurately understand the subject-object relationship of entities. Xu et al. [37] use small models as plug-ins for LLMs to improve model performance on supervised tasks, as well as improve model multilingualism and interpretability.

## 3 PRELIMINARY

In this section, we present the problem formulation for NER. Additionally, we introduce uncertain probabilistic modeling methods [29] and optimization functions [40] to construct local models.

### 3.1 Local Model and Problem Definition

**Local Model**. In this study, we choose SpanNER [10] as the local model for the following reasons:

(1) Web content is noisy and diverse, including variations in writing style, spelling errors, and inconsistent formatting. Span-based models tend to be more robust to such changes because they focus on extracting coherent spans of text rather than labeling individual tokens [40].

(2) Downstream programs such as web retrieval, dialogue, relationship extraction, etc., have high requirements on entity boundaries. Compared with other NER paradigms, SpanNER reduces the ambiguity of boundary expressions [34].

**Problem Definition**. Specifically, Given a sentence of length $n$, denoted as $\mathbf{x} = \{x^1, x^2, ..., x^n\}$. For the commonly used span-based NER modeling method, SpanNER [10] enumerates all candidate spans $S = \{s^1, s^2, ..., s^m\}$, where $m$ represents the set size. $s = \{x^e, x^{e+1}, ..., x^d\}$ represents the entity span between the start $x^e$ and the end $x^d$, and the maximum length of the span within the predetermined range is $l$. Each span is assigned an entity label $c$, where $c$ belongs to the label set $c \in [1, C]$. Among these test entities, if the data distribution is the same as that of the training set, it is called in-domain (ID). Conversely, if the data distribution differs from that of the training set, it is termed OOD. Entities not present in the model's vocabulary are referred to as OOV entities. In general, OOD and OOV entities represent unseen test entities for the model, and the performance reflects the robustness of the model in an open environment.

### 3.2 Uncertainty Estimation Methods

Uncertainty estimation techniques play a key role in deploying machine learning in the field of NLP. Commonly used estimation methods include confidence-based, sampling-based, and distribution-based methods [17]. Recall that SpanNER divides the input into spans, and logits are obtained after feeding each span into the neural network. Based on this, we use the following four uncertainty estimation methods in our study:

- **Least Confidence (LC)**. For a $C-$class classification problem, given an span input $s$ and the class prediction $y \in \{1, ..., C\}$, the uncertainty describe as:

$$u_{LC}(s) = 1 - \max_{c \in C} \ p(y = c|s), \qquad (1)$$

where $p(y = c|s)$ is the Softmax scores on $C$ categories.

- **Prediction Entropy (PE)** serves as a straightforward and efficient method for estimating uncertainty. Maximum entropy occurs when all outcomes share the same probability. The uncertainty $u_{PE}(s)$ of point $s$ can be quantified as its predicted entropy:

$$u_{PE}(s) = \sum_{c \in C} -p(y = c|s) \log p(y = c|s). \qquad (2)$$

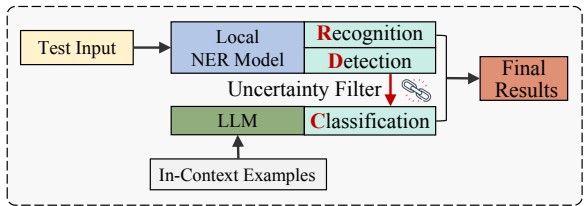

**Figure 2: Link's overall framework. A fine-tuned NER model is used for recognizing entities and detecting uncertain entities, and the LLM reclassifies the detected uncertain entities.**

- **Monte Carlo Dropout (MCD)** is the typical sampling method. Specifically, the MCD method performs $M$ stochastic forward passes with activated dropout:

$$u_{MCD}(s) = \frac{1}{M} \sum_{c,m}^{C,M} -p(y^m = c|s) \log p(y^m = c|s). \qquad (3)$$

In this study, MCD is used to perform Monte Carlo integration on the entropy value.

- **Evidential Neural Network (ENN)** is one of the distribution based uncertainty estimation methods implemented through a Dirichlet distribution parameterized neural network. Specifically, each $s$ is fed into the evidential neural network to obtain *logits* and convert them into evidence $\mathbf{e}$ using a non-negative function: $\mathbf{e} = \mathbf{func}(logits)$. For example, **func** can be an exponential function or Softplus. Finally, the predicted probability $\mathbf{p}$ and the corresponding uncertainty $u_{ENN}$ of the span entity are as follows:

$$\mathbf{p} = \frac{\mathbf{e} + \mathbf{s}^{\text{prior}}}{\sum_C (\mathbf{e} + \mathbf{s}^{\text{prior}})}, \quad u_{ENN}(s) = \frac{\sum_C \mathbf{s}^{\text{prior}}}{\sum_C (\mathbf{e} + \mathbf{s}^{\text{prior}})}, \qquad (4)$$

where $\mathbf{s}^{\text{prior}}$ represents a uniform distribution with a value of $\mathbf{1}$ as the prior.

**Model Learning Optimization**. In addition to the uncertainty quantification method of ENN, other methods use the original loss function of SpanNER, which is the cross-entropy loss function. For ENN, to improve the detection of unseen entities, the training scheme of E-NER [40] is used for network optimization. Overall, the E-NER loss function of this scheme consists of two components: classification loss function $\mathcal{L}_{\text{cls}}$ and the penalty loss function $\mathcal{L}_{\text{penalty}}$. The detailed optimization function is in Appendix B.

$$\mathcal{L}_E = \mathcal{L}_{\text{cls}} + \mathcal{L}_{\text{penalty}}, \qquad (5)$$

where $\mathcal{L}_{\text{cls}}$ allows the evidence to gather on the correct category, and $\mathcal{L}_{\text{penalty}}$ aims to reduce the evidence of the incorrect category, thereby increasing the uncertainty of the unknown entity.

Finally, the tuple $y$ representing each entity prediction is denoted as $y = [(x^e, x^d, \text{label}), u]$.

## 4 LINKNER FRAMEWORK

As shown in Figure 2, we describe the LinkNER overall framework. In the subsequent subsection, we offer a comprehensive explanation of the showcased concept.

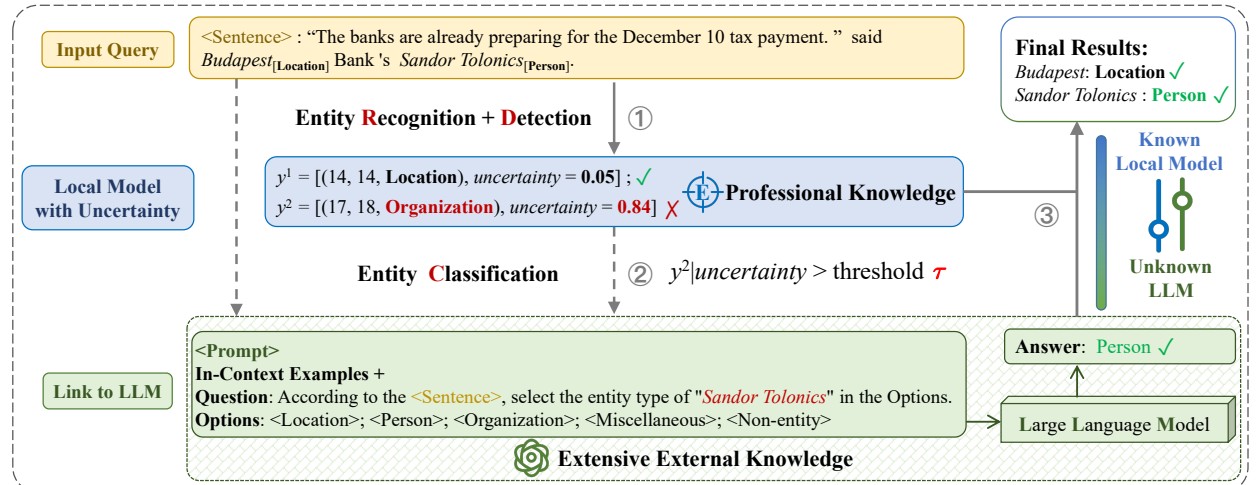

**Figure 3: Illustration of the LinkNER framework and components.**

## 4.1 LinkNER Workflow

The entity recognition of each text contains three steps, and the workflow of LinkNER is shown in Figure 3. At **step** ①, the test sentence is fed into the local model, which outputs the entity recognition result and the corresponding uncertainty probability, denoted as $y = [(x^e, x^d, \text{label}), u]$. At **step** ②, the uncertainty of the local model output is compared with a predefined threshold $\tau$. Entities that are greater than the threshold $\tau$ are detected as uncertain, and sent to the LLM for further classification. Entities that are less than the threshold $\tau$ do not need to undergo refinement. Finally, at **step** ③, the outputs of the local model and LLM are integrated into the final result. Then, we explain in detail how the local model performs entity recognition and detection, and how the LLM refines the results from the local model.

## 4.2 Local NER Model

As mentioned in §3, the local NER model is SpanNER, which is commonly used for NER tasks and comes equipped with four uncertainty estimation methods. These four uncertainty estimation methods are chosen because they cover common uncertainty quantification methods in NLP [17] (based on confidence, sampling, and distribution), as well as targeted quantification methods in NER (E-NER). Here a research question raises: *why the uncertainty needs to be reliable?* The reason relies on that uncertainty needs to accurately characterize the output reliability of the model. For example, using only the output confidence of Softmax often leads to overconfidence [13]. It fails to clearly distinguish between seen and unseen uncertain entities [40]. Sampling-based uncertainty estimation methods consume more time, computing resources, and distribution-based methods typically necessitate model retraining. There is no certainty that a single uncertainty estimation method exhibits the best overall performance across multiple datasets. Therefore, we need to assess the performance of uncertainty methods on various datasets.

After fine-tuning the local model, we can get the relationship between the output uncertainty and model performance. As shown in Figure 4(a), we obtain the performance and entity density of local

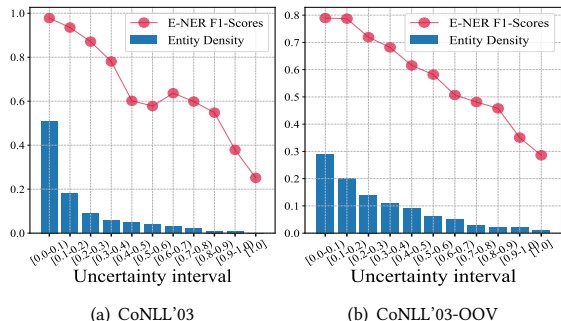

(a) CoNLL'03          (b) CoNLL'03-OOV

**Figure 4: Illustration of local model (E-NER as an example) performance under different uncertainties. The polyline and histogram represent the changes in F1 score and entity density under different uncertainty intervals, respectively.**

model (E-NER as example) under different uncertainty intervals during the CoNLL'03 test. We observe that as uncertainty increases, the performance of the local model and the entity density exhibit a downward trend within each uncertainty interval. This trend indicates that different uncertainty intervals reflect the difficulty of entities. Furthermore, in the OOV test illustrated in Figure 4(b), compared to Figure 4(a), the overall performance experiences a downward shift, and the entity density exhibits a bias towards high uncertainty. These findings demonstrate that the local model, although challenging in OOV testing, still offers opportunities to detect unseen entities through uncertainty.

## 4.3 Linking Local Model to LLM

Entity naming is related to the data sources and annotators involved, which requires the model to carefully capture the data distribution of the current NER task. However, recall our investigation in §1, and previous studies [16] showing that LLM *lack of specialty* (Extract unmentioned entities, undefined categories, etc.) in NER tasks,

hindering LLM from directly extracting entities from a given text. Therefore, we propose an uncertainty-based interaction strategy between two models: Recognition-Detection-Classification (RDC) to address this challenge.

Specifically, a fine-tuned local model is used for entity *recognition*, and its output uncertainty probabilities can be used for uncertain entity *detection*, and then sends uncertain entities to LLM for entity type *classification*, i.e., a multiple-choice question. By doing so, the local model provides LLM with entity span and category constraints, and simultaneously, makes the task paradigm simpler. For clarity, Figure 3 illustrates the process. The uncertain entity *Sandor Tolonics* undergoes filtration using the uncertainty output from the local model. Subsequently, the filtered entity is forwarded to LLM for reclassification. Finally, the output from LLM is combined with the local model's output to obtain the final prediction.

### 4.4 In-Context Learning for LLM

The large and undisclosed parameters of LLM make it difficult to achieve fine-tuning, but its powerful contextual understanding ability makes it possible to perform in-context learning. In order to provide LLM with task-specific information, we design two kinds of context hinting for LinkNER as its components.

**Label Description**. Several datasets already encompass descriptions for different entity types. We argue that integrating these interpretations into LLM augments its semantic comprehension. For instance, the OntoNotes 5.0 [36] dataset offers an explanation for the "FACILITY" entity class: "FACILITY: buildings, airports, highways, bridges, etc." . In the Appendix C Table 8, we furnish label descriptions extracted from the original paper for the dataset.

**Few-Shot Learning**. Examples of each entity category serve as valuable references in this context. We construct an example set by combining the training and validation sets, which comprise <**Question-Answer**> pairs. Each pair consists of context, instructions for entity classification, and answers. To clarify, the sample **Question** and **Answer** format are the same as the question shown in Figure 3 <Prompt>. The question is presented in a multiple-choice format, such as "*select the entity type of Sandor Tolonics*", while the answer is provided by selecting one of the entity type options, such as "Person". Then, we select a total of $N$ classes ($N$-way), and each class has $K$ ($K$-shot) examples from the example set, thus $N \times K$ shots in total. Furthermore, more specific examples can be found in Appendix C. By employing this approach, the few-shot setting equips LLM with a collection of examples for entity classification, allowing the LLM to efficiently classify similar entity types.

## 5 EXPERIMENTS

In this section, we first introduce the benchmark and uncertainty threshold selection, and then conduct extensive experiments to verify the effectiveness of LinkNER.

### 5.1 Experimental Research Questions

In this section, we design extensive experiments to verify the effectiveness of the LinkNER method and the applicable scenarios of the components. We evaluate LinkNER in the following research questions (RQ):

**RQ1**: When should the LLM make decisions in LinkNER?

### Table 2: Statistics of entities in the test sets.

| Dataset | Entities | Types | Domain | OOV Rate |
|---|---|---|---|---|
| CoNLL'03 | 35.1k | 4 | Newswire | - |
| OntoNotes 5.0 | 161.8k | 18 | General | - |
| WikiGold | 1.6k | 4 | General | - |
| CoNLL'03-Typos | 4.1k | 4 | Newswire | 0.71 |
| CoNLL'03-OOV | 5.6k | 4 | Newswire | 0.96 |
| JNLPBA | 4.3k | 5 | Medical | 0.77 |
| WNUT'17 | 0.9k | 6 | SocialMedia | 1.00 |
| TweetNER | 4.0k | 4 | SocialMedia | 0.62 |

**RQ2**: What is the impact of RDC on LinkNER performance?
**RQ3**: How many decisions are made by LLM in LinkNER?
**RQ4**: When is it effective to employ in-context learning in LinkNER?

### 5.2 Experimental Settings

**Dataset**. For testing on standard data, as shown in Table 2, we choose three commonly used datasets, including two In-Domain (ID) datasets and one Out-of-Domain (OOD) dataset: **ID**: CoNLL'03 Dataset [32]. We only consider the English (EN) dataset collected from the Reuters Corpus. **ID**: OntoNotes 5.0 Dataset (Onto. 5.0) [36] has multiple entity types in the fields of telephone conversations, web data, news agencies, etc. **OOD**: WikiGold Dataset [3] is a test set from Wikipedia with the same entity types as CoNLL'03 but with different domains. Furthermore, as shown in Table 2, we use five commonly used OOV datasets to test on unseen entities: Typos Dataset [35] replaces entities in the CoNLL'03 test set by typos version (character modify, insert, and delete operation); OOV Dataset [35] replaces entities in the CoNLL'03 test set for other OOV entities; TweetNER Dataset [38] is a NER dataset derived from English tweets; WNUT'17 Dataset [7], a focus on the noisy and unseen different distribution entities in the social domain. JNLPBA Dataset [5] focuses on the field of biology and contains relevant technical terms and symbols.

To address RQ1, we employ the CoNLL'03 dataset as an example to establish the uncertainty threshold for LLM decision-making. For RQ2 and RQ3, we utilize all datasets (Table 2). For RQ4 scenario, we select CoNLL'03 and its variants, OntoNotes 5.0, and WNUT'17 datasets for ablation experiments and analysis. Furthermore, for RQ1 and RQ4, we use E-NER as the local model uncertainty estimation method, and GPT-3.5 as LLM as an example for analysis. The local model in other scenarios is equipped with all uncertainty estimation methods. Llama 2-Chat (13B) and GPT-3.5 as LLM.

**Benchmark**. To evaluate the effectiveness of LinkNER, we compare it with the following baselines: **SpanNER** [10], which enumerates all possible entities in a sentence, is trained with an unconstrained classification framework. **MINER** [34], which uses variational information bottleneck to solve the OOV problem and demonstrates SOTA performance across various datasets. **DataAug** [6], which trains the architecture using the SpanNER model, and uses the original training set and the entity replacement training set for data augmentation. **VaniIB** [2], which applies the information bottleneck constraint to SpanNER and directly compresses all information in the input.

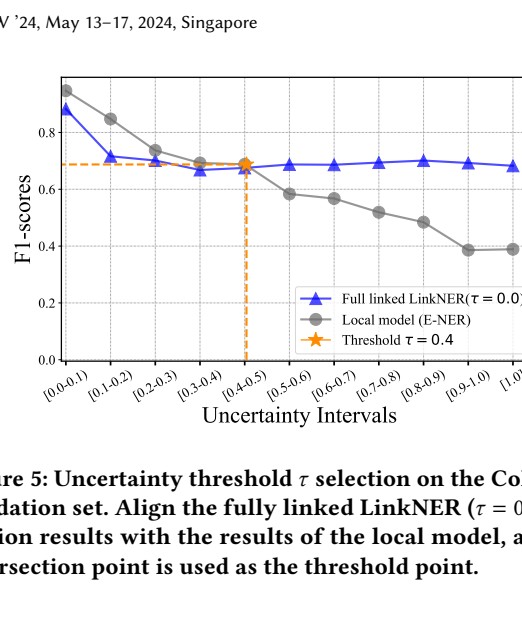

Figure 5: Uncertainty threshold $\tau$ selection on the CoNLL'03 validation set. Align the fully linked LinkNER ($\tau = 0.0$) prediction results with the results of the local model, and the intersection point is used as the threshold point.

**Metrics**. Entity-level micro average F1 score is used for test set evaluation, and only predictions with correct entity boundaries and classifications are considered correct.

## 5.3 Determining Decision Points for LLM in LinkNER

LLM's classification decision depends on the uncertainty estimate score of the local model's output for the entity. The crucial step in LinkNER is the selection of an appropriate uncertainty threshold $\tau$, which determines when LLM makes classification decisions. As mentioned earlier, the performance of the local model deteriorates in the high uncertainty range. Therefore, we aim to leverage LLM for improved entity classification within this range compared to local model. To illustrate this, let us consider CoNLL'03 as an example, as depicted in Figure 5. Initially, we set the LinkNER threshold $\tau = 0.0$ (see full linked LinkNER ($\tau = 0.0$)), as a result, the local model is responsible for entity detection in LinkNER, while LLM (GPT-3.5 is used for this example) provides all entity classification results. Simultaneously, we record the **recognition results of the local model** across various intervals (see E-NER). The threshold $\tau$ is determined by identifying **the intersection point** where the **fully linked LinkNER** aligns with these results (see the star-shaped intersection). In other words, the LLM outperforms the local model in classifying uncertain entities when the uncertainty surpasses the threshold $\tau$. At this moment, integrating the local model with the LLM can lead to performance enhancement. By selecting an appropriate threshold $\tau$, we achieve a favorable balance between performance and efficiency.

Then, we assess the appropriateness of the uncertainty threshold selection. Figures 6(a) and 6(b) present the results, where we employ 10 threshold points with a step size of 0.1. We evaluate the performance of LinkNER at each threshold point individually. The figure displays the outcomes, indicating that both the CoNLL'03 and OOV test sets exhibit the highest F1 scores when the threshold is set to 0.4. Notably, this aligns with the threshold selected in §5.4, based on the CoNLL'03 validation set, which also confirms the rationality of our threshold selection method. This approach

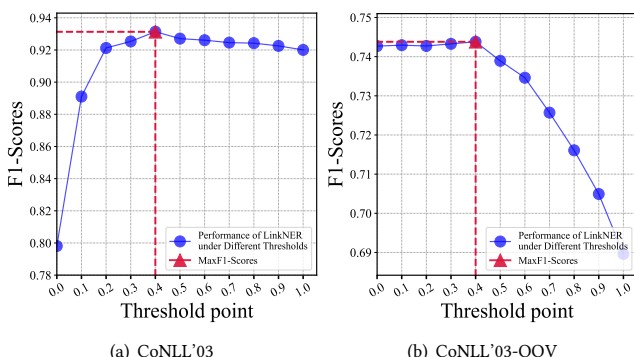

(a) CoNLL'03    (b) CoNLL'03-OOV

Figure 6: Illustrations (a) and (b) present the variations in LinkNER's performance at different threshold points.

allows the local model and the LLM to complement each other. The more serious the phenomenon of unseen entities in the test set, the more serious the *"lack of knowledge"* phenomenon of the local model, so the uncertainty threshold $\tau$ used will be smaller, and the classification of entities is more inclined to LLM decision-making. In this way, we can select the threshold and obtain the performance of the local model and LLM under different uncertainty intervals in more detail. The thresholds for all datasets in the experiments are given in Appendix C Table 7.

## 5.4 Assessing RDC's Influence on LinkNER Performance

**Standard Performance Evaluation**. After the optimal threshold is chosen, we first investigate *how LinkNER performs on the standard evaluation.* After selecting the optimal threshold, we first study the performance of LinkNER on the ID dataset. The experimental results are displayed in the first two columns of Table 3. We confirm that LinkNER generally improves task performance on ID data, demonstrating that local models and LLM can complement each other through the RDC model bridge. Among these models, the minimum and maximum improvements of different local models are relatively close. MCD and E-NER serve as better methods for estimating local model uncertainty. Whether it's Llama 2-Chat (13B) or GPT-3.5 for classification decisions, after linking, the performance of LinkNER closely matches that of the SOTA method. However, MINER, which excels in OOV scenarios, exhibits weaker performance in ID tests. Since the local model acquires relevant knowledge from the LLM, most simple entities are accurately recognized in regular tests, while only a few challenging entities with higher uncertainty are identified and classified by the LLM.

**OOV/OOD Robustness Test Evaluation**. Then, it is wondered whether LinkNER can achieve better results in the robustness test. As shown in the results in columns 3-8 in Table 3. The local model captures the original data distribution, allowing the RDC model to exhibit more powerful uncertain entity recognition capabilities while maintaining high efficiency. Especially in the OOD test set, LinkNER equipped with E-NER and GPT-3.5 exceeds the SOTA method by a **2.87%** F1 score. In the two OOV variant test sets of CoNLL'03, LinkNER equipped with different uncertainty estimation

Table 3: Comparison of performance and robustness tests between LinkNER and SOTA methods, in terms of F1 (%).

| | Setting | Onto. 5.0 ID | CoNLL'03 ID | Typos | OOV | OOD | WNUT'17 OOV | TweetNER OOV | JNLPBA OOV |
|---|---|---|---|---|---|---|---|---|---|
| | **SOTA** | 90.50[24] | 94.54[22] | 87.57[34] | 79.15[34] | 82.21[40] | 54.86[34] | 76.71[40] | 78.47[40] |
| Baseline | GPT-3.5 | 51.15 | 67.08 | 62.24 | 62.74 | 71.28 | 42.53 | 55.59 | 41.25 |
| | VaniIB | - | - | 83.49 | 70.12 | - | 51.60 | 71.19 | 73.41 |
| | DataAug | - | - | 81.73 | 69.60 | - | 52.29 | 73.69 | 75.85 |
| | SpanNER (BERT_large) | 87.82 | 92.37 | 81.83 | 64.43 | 80.54 | 51.83 | 74.93 | 75.07 |
| | SpanNER-MCD (BERT_large) | 88.42 | 92.67 | 83.58 | 69.49 | 78.30 | 49.07 | 75.25 | 76.39 |
| | MINER (RoBERTa_large) | 88.99 | 90.19 | **87.57** | **79.15** | 77.01 | **54.86** | 75.38 | 76.43 |
| | E-NER (BERT_base) | 88.70 | 92.00 | 82.69 | 68.61 | 81.09 | 51.74 | 75.64 | 76.44 |
| | E-NER (BERT_large) | **90.59** | **93.12** | 84.62 | 70.41 | **82.21** | 51.28 | **76.71** | **78.47** |
| Link-Llama2 (13B) | Link-SpanNER (Confidence) | 89.14 | 93.04 | 83.83 | 70.21 | 82.40 | 62.11 | 77.51 | 78.09 |
| | Link-SpanNER (Entropy) | 89.24 | 93.28 | 83.87 | 70.56 | 82.65 | 62.81 | 77.20 | **82.89** |
| | Link-SpanNER (MCD) | 90.06 | **93.99** | **85.16** | 71.41 | 82.42 | 63.38 | **77.77** | 79.85 |
| | Link-ENER (BERT_large) | **90.64** | 93.17 | 85.03 | **73.12** | **82.72** | **68.30** | 77.05 | 80.41 |
| | Min Δ LinkNER vs. LocalNER | 0.05↑ | 0.05↑ | 0.41↑ | 1.92↑ | 0.51↑ | 10.28↑ | 0.34↑ | 1.94↑ |
| | Max Δ LinkNER vs. LocalNER | 1.64↑ | 1.32↑ | 2.04↑ | 6.13↑ | 4.12↑ | 17.02↑ | 2.58↑ | 7.82↑ |
| Link-GPT3.5 | Link-SpanNER (Confidence) | 89.20 | 92.99 | 83.85 | 69.63 | 82.51 | 65.12 | 79.18 | 82.22 |
| | Link-SpanNER (Entropy) | 88.72 | 93.32 | 84.41 | 71.24 | 83.79 | 65.45 | 79.20 | 82.89 |
| | Link-SpanNER (MCD) | 90.45 | **94.18** | 85.48 | 71.95 | 82.49 | 66.67 | **80.12** | 81.93 |
| | Link-ENER (BERT_base) | 89.71 | 93.05 | 85.73 | **72.91** | **85.08** | **73.04** | 77.02 | 84.60 |
| | Link-ENER (BERT_large) | **90.82** | 93.36 | **86.52** | 73.74 | 84.16 | 72.12 | 79.43 | **86.81** |
| | Min Δ LinkNER vs. GPT-3.5 | 37.75↑ | 25.91↑ | 21.61↑ | 6.89↑ | 11.21↑ | 22.59↑ | 21.43↑ | 40.68↑ |
| | Max Δ LinkNER vs. GPT-3.5 | 39.67↑ | 27.10↑ | 24.28↑ | 11.99↑ | 13.80↑ | 30.51↑ | 24.53↑ | 45.56↑ |
| | Min Δ LinkNER vs. LocalNER | 0.18↑ | 0.24↑ | 1.90↑ | 2.46↑ | 1.95↑ | 13.29↑ | 1.38↑ | 5.54↑ |
| | Max Δ LinkNER vs. LocalNER | 2.03↑ | 1.51↑ | 3.04↑ | 6.81↑ | 4.19↑ | 21.30↑ | 4.87↑ | 8.34↑ |

methods shows improved performance in terms of F1 scores. The overall performance of MCD and E-NER is better, but it is still far from SOTA. The gap, we analyze, is related to the choice of the local model. The experiments we conduct only in the original framework of SpanNER do not include any information enhancement. It is worth noting that in OOV/OOD scenarios, different uncertainty estimation methods have a greater impact on the final performance of LinkNER, which is manifested in the floating gap between the minimum and maximum performance improvements. These findings further validate the usability and robustness of the LinkNER model equipped with RDC strategy.

Furthermore, LinkNER outperforms SOTA in F1 scores on challenging social media datasets, including TweetNER and WNUT'17 datasets. These datasets contain a lot of noise and unseen entities, especially LinkNER's performance improvement of **18.18%** on the WNUT'17 dataset relative to SOTA, which proves the reliability of LinkNER even on unstructured social media noisy data. Finally, in the medical dataset JNLPBA, the entity test contains OOV entities with a large number of biological terms. LinkNER still surpass the SOTA with an **8.34%** F1 score, proving the effectiveness of LinkNER in NER tasks in the medical professional field.

## 5.5 Analyzing the Uncertainty Interval of LLM Decision-making

To further investigate the contribution of LLM in LinkNER, we analyze the entity recognition performance across various uncertainty intervals. As illustrated in Figure 7, LinkNER can improve

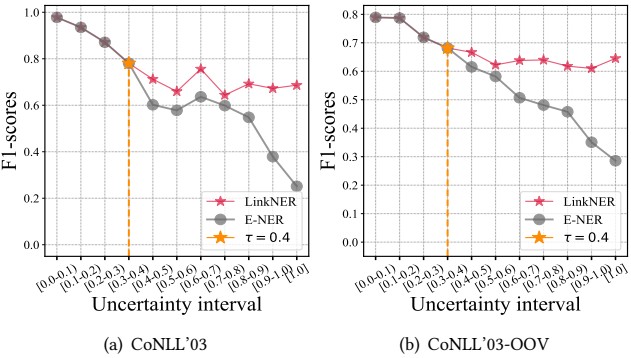

(a) CoNLL'03  (b) CoNLL'03-OOV

Figure 7: Illustrations (a) and (b) depict the performance comparison between LinkNER and E-NER across various uncertainty intervals.

performance in high uncertainty intervals. Recall that Section §3 mentions that entities with high uncertainty are more likely to be incorrect, and a reasonable threshold link LLM can compensate for the test performance in the high uncertainty interval. Consequently, LinkNER can improve the overall performance by refining high-uncertainty entities, whether it is regular data testing 7(a) or testing with unseen entities 7(b). This further verifies the rationality of LinkNER to achieve more robust performance, ensuring that the system is still available under noise data.

**Table 4: The proportion of LLM making classification decisions in different LinkNERs.**

| Setting | Onto. 5.0 ID | CoNLL'03 | | | WNUT'17 | TweetNER | JNLPBA |
| | | ID | Typos | OOV | OOD | OOV | OOV | OOV |
|---|---|---|---|---|---|---|---|---|
| Link-SpanNER (Confidence) | 0.47% | 0.08% | 0.27% | 1.18% | 0.49% | 59.70% | 1.74% | 21.58% |
| Link-SpanNER (Entropy) | 4.59% | 2.23% | 4.09% | 6.88% | 8.13% | 60.18% | 10.41% | 45.31% |
| Link-SpanNER (MCD) | 0.16% | 0.44% | 0.95% | 0.35% | 0.66% | 13.82% | 0.36% | 6.33% |
| Link-ENER (BERT_base) | 1.56% | 2.81% | 12.90% | 21.17% | 19.79% | 85.13% | 72.65% | 50.35% |
| Link-ENER (BERT_large) | 0.94% | 2.38% | 9.29% | 13.78% | 14.33% | 55.66% | 9.45% | 52.58% |

Furthermore, we analyze the proportion of LLM making classification decisions. The results are shown in Table 4, from the perspective of the dataset, the four uncertainty methods screen out a relatively small number of uncertain entities on the ID dataset, which aligns with our hypothesis. There are only a small proportion of challenging entities in these ID datasets. However, it filters out more uncertain entities in the WNUT'17 dataset. We find that this is caused by too little data available in the training set and underfitting of the model. Although the uncertainty estimation method based on entropy value filters out more uncertain entities, the improvement is subtle and consumes too much LLM reasoning resources. Table 3 shows that the final performance improvement is very limited. Conversely, the method based on MCD screens out fewer uncertain entities, but Table 3 shows that the final performance is greatly improved. While MCD multiple forward sampling consumes some local model reasoning resources, high-quality uncertainty estimation yields more effective results for LLM. E-NER detects more uncertain entities (as demonstrated by BERT_base), and performance improves noticeably in OOV/OOD scenarios. However, overly "cautious" uncertainty scores also consume more LLM reasoning resources.

## 5.6 Identifying Effective Scenarios for In-Context Learning in LinkNER

We conduct ablation experiments to assess the effect of model components and in-context learning in LinkNER. The model components considered are as follows: (I) Local model E-NER, which focuses on entity recognition and detection in the first stage; (II) The GPT-3.5 plugin, responsible for making decisions regarding uncertain entities. Furthermore, we perform a study on the in-context learning component of LinkNER, exploring the following aspects: (a) The multiple-choice-style few-shot setting. (b) The information of entity type label description. The experimental results are shown in Table 5. When compared to the local model alone (E-NER), LinkNER improves the F1 score of the ID datasets by 1.05%↑ and 1.01%↑, respectively. For the Typos and OOV variant datasets of ConLL'03 and the WNUT'17 dataset, LinkNER's F1 scores improve by 3.04%↑, 4.30%↑, and 21.30%↑, respectively. Additionally, when compared to GPT-3.5, the F1 scores of LinkNER on ID data are 25.97%↑ and 38.56%↑, respectively. LinkNER achieves improvements of 23.49%↑, 10.17%↑, and 30.51%↑ on the Typos, OOV variant dataset, and WNUT'17 dataset, respectively.

For each component, we analyze CoNLL'03 and its variants, as well as the Onto. 5.0 dataset, this can simulate normal text and a small amount of noise text on the web. For setting (1), when all

**Table 5: Ablation study evaluation results F1(%) . (a) Few-shot setting (FS). (b) Entity type label description (LD).**

| Components | | CoNLL'03 | | | Onto. | WNUT. |
| (a) *FS.* | (b) *LD.* | ID | Typos | OOV | ID | OOV |
|---|---|---|---|---|---|---|
| (I) E-NER (BERT_base) | | 92.00 | 82.69 | 68.61 | 88.70 | 51.74 |
| (II) GPT-3.5 | | 67.08 | 62.24 | 62.74 | 51.15 | 42.53 |
| (1) ✗ | ✗ | 92.50 | 83.60 | 71.25 | 89.16 | **77.96** |
| (2) ✗ | ✓ | 92.85 | 85.51 | 70.68 | 89.59 | 77.54 |
| (3) ✓ | ✗ | 92.94 | 84.79 | 71.60 | 89.37 | 64.51 |
| ✓ | ✓ | **93.05** | **85.73** | **72.91** | **89.71** | 73.04 |

examples containing context are removed, both models work seamlessly together without any adverse impact. However, relative to LinkNER's performance, there is a sharp decline. RDC's interaction model fully leverages the respective benefits of both models. For the setup of the LD (2), we attempt to remove the FS setting from LinkNER, resulting in performance degradation. This degradation can be attributed to the lack of examples for each entity type, making it difficult for GPT-3.5 to capture the data distribution. (3) When the LD is removed, performance also decreases. This demonstrates that accurate entity LD provides a classification basis for GPT-3.5. However, for the WNUT'17 dataset, which is often noisy data in web social media, we observe that contextual examples and LD hinder performance improvement. We attribute this to the limited size of the training data and resulting in a significant distribution gap between the training and test sets. Consequently, when the training data distribution matches that of the web test text, we can use context learning to enhance the effectiveness of LLM in making classification decisions, and vice versa.

## 6 CONCLUSION

In this work, we combine LLM to design a more robust NER system by utilizing the local fine-tuned model, SpanNER. To address the issue of *"lack of knowledge"* in the fine-tuning model for unseen entities, as well as the *"lack of specialty"* problem of LLM entity recognition, we propose the LinkNER framework, which we refer to as the RDC linking strategy based on uncertainty estimation, to link these two models. Extensive experimental results demonstrate that the proposed method can be effectively applied to benchmarks and robustness tests, particularly outperforming the SOTA models by 3.04% to 21.30% F1 scores in the challenging robustness tests. These results validate the superiority of LinkNER in open environments.

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

## A    CASE STUDY

We conduct a case study to better demonstrate the working behavior of LinkNER. Examples are given in Table 6. In the social media data examples, the local model encounters difficulty in recognizing the entities "*stephenjeske*" and "*CBC Manitoba*". However, through the RDC strategy, the LinkNER can correctly classify these two entity categories. We find that the effectiveness of GPT-3.5 also depends on the recognition and detection results of the local model. Therefore, the uncertainty-based RDC strategy takes advantage of the two models in LinkNER and makes them complement each other.

## B    E-NER OPTIMIZATION FUNCTION

In this section, we give the detailed formulation of the E-NER optimization function. E-NER mainly includes two components, namely the classification loss function $\mathcal{L}_{\text{cls}}$ and the penalty loss function $\mathcal{L}_{\text{penalty}}$, as follows:

$$\mathcal{L}_E = \mathcal{L}_{\text{cls}} + \mathcal{L}_{\text{penalty}}. \tag{6}$$

Specifically, given a sample $(x^i, \mathbf{y}^i)$, the two components are as follows:

**(a)** training utilizes cross-entropy loss to learn evidence for the correct class:

$$\mathcal{L}_{\text{cls}}^i = \sum_{c=1}^{C} W_c^i \left( \psi(S^i) - \psi(\alpha_c^i) \right), \tag{7}$$

where $W^i = (1 - \mathbf{e}^i/\alpha_0) \odot \mathbf{y}^i$ is reweighed one-hot $C$-dimensional label for sample $x^i$ to reduce overfitting, $\psi(\cdot)$ is the digamma function.

**(b)** Penalty terms include KL divergence and uncertainty optimization terms. KL divergence serves as a distribution penalty for other classes, and uncertainty optimization focuses on wrong entities:

$$\mathcal{L}_{\text{penalty}}^i = \lambda_1 KL[\text{Dir}(\mathbf{p}^i | \widetilde{\boldsymbol{\alpha}}^i) || \text{Dir}(\mathbf{p}^i | \mathbf{1})] - \\ \lambda_2 \sum_{i \in \{\hat{y}^i \neq y^i\}} \log(u^i), \tag{8}$$

where $\lambda_1$ and $\lambda_2$ are the balance factor, $\text{Dir}(\mathbf{p}^i | \mathbf{1})$ is a special case which is equivalent to the uniform distribution, and $\widetilde{\boldsymbol{\alpha}}^i = \mathbf{y}^i + (1 - \mathbf{y}^i) \odot \boldsymbol{\alpha}^i$ denotes the masked parameters while $\odot$ refers to the Hadamard (element-wise) product, which removes the non-misleading evidence from predicted parameters $\boldsymbol{\alpha}^i$.

## C    IMPLEMENTATION DETAILS

For local models, we use E-NER with BERT-base-uncased as the base encoder [8]. The dropout rate is set to 0.2 and the BERT dropout rate is set to 0.15. The AdamW optimizer [26] is used for training

the CoNLL'03 dataset with a learning rate of 1e-5 and a training batch size of 10. To improve training efficiency, sentences are truncated to a maximum length of 128, and the maximum length of the span enumeration is set to 4. The initial value of $\lambda_0$ is set to 1e-02. We use heuristic decoding and retain the highest probability span for flattened entity recognition in span-based methods. In this study, the large language model chose gpt-3.5-turbo and Llama 2-Chat as the linked LLM. As shown in Table 7, we decide the uncertainty threshold $\tau$ by the method in §5.4. It is worth noting that CoNLL'03-Typos, CoNLL'03-OOV and WikiGold all use CoNLL'03 as the source dataset, so the label description and threshold $\tau$ also use the corresponding information of the CoNLL'03 validation set. Moreover, we summarize the label descriptions in Table 8.

**Table 6: A comprehensive case study of LinkNER on NER tasks in real-world scenarios. It includes two parts, one is the example construction of LLM in-context learning, and the other is the LinkNER workflow.**

| | |
|---|---|
| **in-context Learning** | <Label Description>
Here is all entity class information: Person: Names of people (e.g. Virginia Wade)......

<Few-Shot (6-way, 1-shot) examples>
Example 1: "<Context>:*Pxleyes* Top 50 Photography Contest Pictures of August 2010 ... http://bit.ly/bgCyZ0 #photography Select the entity type of *Pxleyes* in this context, and only need to output the entity type."
Answer: corporation
...
Example 6: "<Context>:@SnoopDogg hey snoop my wife Cath is 30 today , any chance of a shout out to her , Select the entity type of *snoop* in this context, and only need to output the entity type."
Answer: person |
| **LinkNER's Work Case** | <Context>
**1.** "<Context>: 9 Resources For Crafting The Perfect Outreach Email by @ *stephenjeske* via @ quora https://t.co/c66x410IUr # emailmarketing # startup"
**2.** "<Context>: You do realize this was published by *CBC Manitoba* ."
**3.** "<Context>: That seems like something someone who supposedly works in *Sandringham* "

<Local Model>
**1.** stephenjeske$_{\{10,10,O|u=0.8\}}$ ✗
**2.** CBC Manitoba$_{\{7,8,location|u=0.9\}}$ ✗
**3.** Sandringham$_{\{10,10,location|u=0.0\}}$ ✓

<Prompt and Respond>
**Question 1.** Label description + Few-Shot examples + "<Context>: 9 Resources For Crafting The Perfect Outreach Email by @ *stephenjeske* via @ quora https://t.co/c66x410IUr # emailmarketing # startup". Select the entity type of *stephenjeske* in this context, and only need to output the entity type: location; group; corporation; person; creative-work; product; Non-entity.
**Answer:** person
**Question 2.** Label description + Few-Shot examples + "<Context>: You do realize this was published by CBC Manitoba ." Select the entity type of *CBC Manitoba* in this context, and only need to output the entity type: location; group; corporation; person; creative-work; product; Non-entity.
**Answer:** corporation

<Final Results>
**1.** stephenjeske$_{\{10,10,person\}}$ ✓
**2.** CBC Manitoba$_{\{7,8,corporation\}}$ ✓
**3.** Sandringham$_{\{10,10,location\}}$ ✓ |

**Table 7: Threshold settings for each uncertainty method on different datasets.**

| Setting | CoNLL'03 | | | | WNUT'17 | TweetNER | JNLPBA | OntoNotes 5.0 |
|---|---|---|---|---|---|---|---|---|
| | ID | Typos | OOV | OOD | OOV | OOV | OOV | ID |
| Link-SpanNER (Confidence) | 0.4 | 0.4 | 0.4 | 0.4 | 0.1 | 0.3 | 0.4 | 0.8 |
| Link-SpanNER (Entropy) | 0.7 | 0.7 | 0.7 | 0.7 | 0.3 | 0.8 | 0.7 | 0.9 |
| Link-SpanNER (MCD) | 0.8 | 0.8 | 0.8 | 0.8 | 0.1 | 0.6 | 0.4 | 0.7 |
| Link-ENER (BERT_base) | 0.4 | 0.4 | 0.4 | 0.4 | 0.1 | 0.2 | 0.2 | 0.5 |
| Link-ENER (BERT_large) | 0.8 | 0.8 | 0.8 | 0.8 | 0.1 | 0.4 | 0.3 | 0.9 |

**Table 8: The label description information in each dataset involved in the experiment, and the selected threshold.**

| Dataset | Label description |
|---|---|
| CoNLL'03 CoNLL'03-Typos CoNLL'03-OOV WikiGold | "Here is the entity class information: Person: This category includes names of persons, such as individual people or groups of people with personal names. Organization: The organization category consists of names of companies, institutions, or any other group or entity formed for a specific purpose. Location: The location category represents names of geographical places or landmarks, such as cities, countries, rivers, or mountains. Miscellaneous: The miscellaneous category encompasses entities that do not fall into the above three categories. This includes adjectives, like Italian, and events, like 1000 Lakes Rally, making it a very diverse category." |
| OntoNotes 5.0 | "Here is the entity class information: PERSON: People, including fictional; ORGANIZATION: Companies, agencies, institutions, etc. GPE: Countries, cities, states; DATE: Absolute or relative dates or periods; NORP: Nationalities or religious or political groups; CARDINAL: Numerals that do not fall under another type; TIME: Times smaller than a day; LOC: Non-GPE locations, mountain ranges, bodies of water; FACILITY:Buildings, airports, highways, bridges, etc; PRODUCT: Vehicles, weapons, foods, etc. (Not services); WORK_OF_ART: Titles of books, songs, etc; MONEY: Monetary values, including unit; ORDINAL: "first", "second", etc; QUANTITY: Measurements, as of weight or distance, etc; EVENT: Named hurricanes, battles, wars, sports events, etc; PERCENT: Percentage (including "%"), etc; LAW: Named documents made into laws, etc; LANGUAGE: Any named language, etc." |
| Wnut'17 | "Here is all entity class information: Person: Names of people (e.g. Virginia Wade). Don't mark people that don't have their own name. Include punctuation in the middle of names. Fictional people can be included, as long as they're referred to by name (e.g. Harry Potter). Location: Names that are locations (e.g. France). Don't mark locations that don't have their own name. Include punctuation in the middle of names. Fictional locations can be included, as long as they're referred to by name (e.g. Hogwarts). Corporation: Names of corporations (e.g. Google). Don't mark locations that don't have their own name. Include punctuation in the middle of names. Product: Name of products (e.g. iPhone). Don't mark products that don't have their own name. Include punctuation in the middle of names. Fictional products can be included, as long as they're referred to by name (e.g. Everlasting Gobstopper). It's got to be something you can touch, and it's got to be the official name. Creative-work: Names of creative works (e.g. Bohemian Rhapsody). Include punctuation in the middle of names. The work should be created by a human, and referred to by its specific name. Group: Names of groups (e.g. Nirvana, San Diego Padres). Don't mark groups that don't have a specific, unique name, or companies (which should be marked corporation)." |
| TweetNER | "Here is the entity class information: For polysemous entities, our guidelines instructed annotators to assign the entity class that corresponds to the correct entity class in the given context. For example, in "We're driving to Manchester", Manchester is a Location, but in "Manchester are in the final tonight", it is a sports club – an Organization. Special attention is given to username mentions. Where other corpora have blocked these out or classified them universally as Person, our approach is to treat these as named entities of any potential class. For example, the account belonging to the "Manchester United football club" would be labeled as an Organization. Other: The Other category encompasses entities that do not fall into the above three categories. This includes adjectives, like Italian, and events, like 1000 Lakes Rally, making it a very diverse category." |
| JNLPBA | "Here is the entity class information: The composition of the Protein category includes protein molecules, complexes, etc. The DNA is Deoxyribonucleic Acid. The RNA is Ribonucleic Acid. The Cell_type category refers to the Cell type in nature. The Cell_line category refers to the Cell line in artificial." |

