# OpenReview forum: "LinkNER: Linking Local Named Entity Recognition Models to Large Language Models using Uncertainty"
_ACM.org/TheWebConf/2024/Conference — TheWebConf24 Oral_

### Official Review · Reviewer_AWSH · 2023-11-12

**Novelty:** 7
**Technical Quality:** 6

**Review:**

Quality
- **High Quality**: The paper demonstrates high quality in its research design, methodology, and analysis. The experiments are well-structured and adequately support the claims made. The use of both standard NER test sets and challenging datasets like noisy social media data ensures a comprehensive evaluation of the LinkNER framework.
- **Attention to Detail**: The paper provides detailed insights into the integration of fine-tuned models with LLMs, addressing the challenges in NER tasks, especially in recognizing unseen entities.

Clarity
- **Well-Organized and Clear**: The paper is well-organized, with a clear flow from the introduction to the conclusion. Each section logically leads to the next, making it easy for readers to follow the progression of the research.
- **Accessible Language**: The authors use clear and concise language, making complex concepts accessible to a broad audience, including those not specialized in NER or LLMs.

Originality
- **Innovative Approach**: The paper's approach to integrating fine-tuned local models with LLMs for NER tasks is original and innovative. This novel strategy addresses a significant gap in the existing literature.
- **Unique Contribution**: The introduction of the RDC linking strategy based on uncertainty estimation is a unique contribution, setting this work apart from previous studies in the field.

Significance
- **Substantial Impact**: The paper's findings have substantial significance in the field of NER and NLP. By demonstrating improved performance in recognizing unseen entities, the research contributes to enhancing the effectiveness of NER systems in various applications.
- **Broad Relevance**: The relevance of this work extends beyond NER tasks, offering insights that could be applicable to other areas of NLP and AI. It provides a foundation for future research in integrating different types of models for improved performance in complex language processing tasks.

What can be improved:
- Check the values for "Ratio@SOTA" in Table 2, should be 70.95\% for CoNLL'03, 56.52\% for Onto. 5.0 and 52.57\% for JNLPBA;
- Check the value of "Min $\Delta$ LinkNER vs. GPT-3.5" @ Onto 5.0 ID, should be 37.57;
- Check the value of "Max $\Delta$ LinkNER vs. GPT-3.5" @ CoNLL’03 OOV, should be 11.00;
- Correct the improvements over SOTA in the Abstract, you showed Max $\Delta$ LinkNER (LinkGPT3.5) vs. LocalNER instead. Correct numbers for Best LinkNER vs. SOTA should be: CoNLL'03 Typos: -1.05, CoNLL'03 OOV: -5.41, CoNLL'03 OOD: 2.87, WNUT'17 OOV: 18.18, TweetNER OOV: 3.41, JNLPBA OOV: 8.34;
- Add "Best LinkNER vs. SOTA" to Table 3 to align the results to the statement in the Abstract.

Overall Pros:
- Innovative integration of fine-tuned models with LLMs;
- Significant improvement in NER performance, especially in challenging environments;
- Comprehensive experimental validation and analysis.

Overall Cons:
- Limited exploration of the framework's generalizability among domains (e.g., legal, financial, or technical texts) and other languages besides English;
- Heavy reliance on LLMs in some cases, raising sustainability questions;
- Some small technical issues;
- Wrong results in the Abstract and Conclusion: Max $\Delta$ LinkNER (LinkGPT3.5) vs. LocalNER instead of Best LinkNER vs. SOTA.

**Questions:**

- Could you elaborate on the amount of shots (K) leading to the results in Table 3? Is this a fixed amount per model, a fixed amount per dataset or a hyperparameter? How did you find the best value for K?
- How do you envision the long-term sustainability of the LinkNER framework, particularly in the context of the evolving landscape of LLMs?
- In your view, how much does the performance of the framework change in the multilingual environment?

**Ethics Review Description:**

No ethical issues.

**Reviewer Confidence:**

4: The reviewer is certain that the evaluation is correct and very familiar with the relevant literature

**Scope:**

4: The work is relevant to the Web and to the track, and is of broad interest to the community

---

### Official Review · Reviewer_7FBK · 2023-11-23

**Novelty:** 4
**Technical Quality:** 5

**Review:**

This paper focuses on named entity recognition (NER) by combining local NER model with large language model (LLM). The motivation of this paper is that local model performs poorly on unseen entity recognition due to “lack of knowledge”, while LLM possesses extensive external knowledge but expresses “lack specialty” for NER tasks. Therefore, in order to complement each other, the authors propose LinkNER that combines small fine-tuned models with LLMs by an uncertainty-based linking strategy.

Strengths:
1. The named entity recognition (NER) task is important, especially in the era of LLMs, combining small fine-tuned NER models with large language models.
2. Extensive experiments are solid.
3. The writing is easy to follow.

Weaknesses:
1. Some experimental settings are unclear.
2. The efficiency of linking local models to LLM needs further exploration.

**Questions:**

1. OOD is a scenario that this paper focuses on, the corresponding dataset is WikiGold which contains multiple domains. Is there any overlap in the domain between training and testing sets? I don't seem to have seen the experimental results on WikiGold dataset.

2. The proposed framework works that a fine-tuned local model is used for entity recognition, and its output uncertainty probabilities can be used for uncertain entity detection, and then sends uncertain entities to LLM for entity type classification. For OOV or OOD entities, it is also challenging for local model, which leads to some errors for entity spans. However, the subsequent LLM which is only responsible for entity type classification can not correct the span errors?

3. What function was used for ENN in this paper, an exponential function or Softplus?

4. What is the entity density?

5. Line 503-504, what are K and N set to?

6. Figure 5 make me confusing, uncertainty threshold has been set to 0.0, why does the F1 score for Full linked LinkNER still change with the change of Uncertainty Intervals?

7. For Link-SpanNER (Confidence), Link-SpanNER (Entropy) and Link-SpanNER (MCD), what is the local model?

8. How is the performance of naïve Llama2 (13B)?

9. “Hogwarts” was mistakenly written as “Dumbledore” in Figure 1.

**Reviewer Confidence:**

3: The reviewer is confident but not certain that the evaluation is correct

**Scope:**

4: The work is relevant to the Web and to the track, and is of broad interest to the community

---

### Official Review · Reviewer_LtfW · 2023-11-24

**Novelty:** 3
**Technical Quality:** 5

**Review:**

## Overview

In this paper, the authors propose a framework called LinkNER which combines a local NER model with an LLM for the NER task. Uncertainty estimation methods are utilized to decide whether to seek the prediction from the LLM for the final entity type classification. Experiments on multiple datasets show the advantages of the proposed method.


## Strengths of this paper

- The paper is clearly written and the method is simple.
- Experimental results show that existing NER methods can be further improved when equipped with LinkNER, especially for OOV/OOD scenarios.
- On multiple OOV datasets, the proposed method outperforms previous SOTA.

## Weaknesses of this paper

-  The uncertainty threshold seems to be dataset-dependent. This setting can be difficult for real-world applications.
- Since Table 3 already shows the SOTA results, I am wondering whether LinkNER could further improve its performance.
- The proposed method can be expensive since LLM inference usually costs more and has lower latency. Since the authors already know that LLMs can perform better on some uncertainty interval buckets, why not distill the LLM’s ability on those buckets to smaller models ?
- I would like to see the performance of the LLM (i.e. llama 2)  after fine-tuning with NER data.

**Questions:**

see details in the review.

**Reviewer Confidence:**

4: The reviewer is certain that the evaluation is correct and very familiar with the relevant literature

**Scope:**

3: The work is somewhat relevant to the Web and to the track, and is of narrow interest to a sub-community

---

### Decision · Program_Chairs · 2024-01-22

**Decision:**

Accept (Oral)

**Comment:**

Proposes and evaluates a method for named entity recognition that combines a small, fine-tuned model with a black-box LLM. The topic is a reasonable fit with the conference. The paper is very readable. The idea represents sufficient novelty to support publication. The experiments are appropriate and support the conclusions.

 Although there are only three reviews, they are consistent. The authors have engaged with the reviewing process, and in my opinion, have adequately addressed the concerns of the reviewers. I don't see why we wouldn't accept this paper.

 I reccoment poster presentation because I'm not sure this paper will have broad appeal. It's interesting, but the topic is a bit narrow.